# Initial Characterization of 3D Culture of Yolk Sac Tissue

**DOI:** 10.3390/ani13091435

**Published:** 2023-04-22

**Authors:** Vitória Mattos Pereira, Priscila Avelino Ferreira Pinto, Lina Castelo Branco Motta, Matheus F. Almeida, André Furugen Cesar de Andrade, Ana Paula Pinoti Pavaneli, Carlos Eduardo Ambrósio

**Affiliations:** 1Department of Veterinary Medicine, Faculty of Animal Science and Food Engineering (FZEA), University of São Paulo (USP), Pirassununga 13635-900, SP, Brazil; vmattosp@uwyo.edu (V.M.P.);; 2School of Pharmacy, University of Wyoming, Laramie, WY 82072, USA; 3Department of Animal Reproduction, School of Veterinary Medicine and Animal Science (FMVZ), University of São Paulo USP, Pirassununga 13635-900, SP, Brazil

**Keywords:** Matrigel, Hanging-Drop, canine, porcine, bovine, stem cell

## Abstract

**Simple Summary:**

During the early stages of embryonic development, the yolk sac (YS) performs a crucial role in performing hematopoietic, metabolic, and nutritional functions. However, the mechanism of YS transportation and its malfunction leading to yearly miscarriage is not yet clearly understood. In order to address these issues, three-dimensional (3D) culture models of the YS were created and characterized for three different domestic species: canine, bovine, and porcine. Through the utilization of specific culture media, 3D cultures were successfully generated for all three species. Afterwards, the morphology, protein, and mRNA expression related to YS functions were compared. This development represents a significant advancement as it sets the foundation for further investigation of the functional mechanisms of YS tissue.

**Abstract:**

The role of the yolk sac (YS) in miscarriage is not yet clear, largely due to ethical reasons that make in vivo studies difficult to conduct. However, 3D cultures could provide a solution to this problem by enabling cells to be arranged in a way that more closely mimics the structure of the YS as it exists in vivo. In this study, three domestic species (porcine, canine, and bovine) were chosen as models to standardize 3D culture techniques for the YS. Two techniques of 3D culture were chosen: the Matrigel^®^ and Hanging-Drop techniques, and the 2D culture technique was used as a standardized method. The formed structures were initially characterized using scanning electron microscopy (SEM), immunohistochemistry (IHC), and quantitative real-time PCR (RT-qPCR). In general, the 3D culture samples showed better organization of the YS cells compared to 2D cultures. The formed structures from both 3D methods assemble the mesothelial layer of YS tissue. Regarding the IHC assay, all in vitro models were able to express zinc and cholesterol transport markers, although only 3D culture techniques were able to generate structures with different markers pattern, indicating a cell differentiation process when compared to 2D cultures. Regarding mRNA expression, the 3D models had a greater gene expression pattern on the Hemoglobin subunit zeta-like (*HBZ*) gene related to the YS tissue, although no significant expression was found in Alpha-fetoprotein (*AFP*), indicating a lack of endodermal differentiation in our 3D model. With the initial technique and characterization established, the next step is to maintain the cultures and characterize the diversity of cell populations, stemness, functions, and genetic stability of each 3D in vitro model.

## 1. Introduction

The yolk sac (YS) is a crucial placental membrane in mammals. It is responsible for providing nourishment to the developing embryo during the initial first trimester of pregnancy when vascular communication is incomplete [1]. The YS performs a critical role in protein and amino acid transportation and synthesis, and vitamin transport, all of which are essential for early embryonic development [2,3,4]. In addition, the YS serves as the primary site of hematopoiesis in the embryo [5], which produces macrophage progenitors that will populate adult tissues [6]. The YS also contributes to the formation of progenitor germ cells [7].

Although it is well-known that the YS performs a crucial role in embryonic development by providing essential nutrients and serving as the first site of blood cell formation, the relationship between YS abnormalities and first-trimester pregnancy loss is still unclear [3]. Traditional two-dimensional (2D) in vitro culture models are inadequate for studying YS development because they do not accurately mimic the in vivo environment due to the lack of proper cell-to-cell and cell-to-extracellular matrix interactions. In 2D systems, cells tend to be spread out, which does not simulate the tridimensional structure of tissue, impacting many processes, including proliferation, differentiation, and cell death [8].

Therefore, exploring alternative methods for cultivating cells and tissues in vitro is essential. Compared to 2D systems, three-dimensional (3D) cell cultures provide a more precise replication of in vivo conditions, resulting in more reliable experimental results. The 3D microenvironment better mimics in vivo conditions, allowing for a more accurate representation of cell behavior [8]. Additionally, 3D cultures can support the formation of complex cell-to-cell interactions and tissue architectures that are not possible in 2D cultures [9], which is critical for studying the functional mechanisms of YS tissue. Lastly, 3D culture systems allow for the study of drug responses and toxicity in a more realistic and predictive manner, making it beneficial for drug development and testing, and understanding the potential effects of environmental toxins on developing embryos [8,9].

For this study, three different species of domestic animals (porcine, bovine, and canine) were chosen as models due to their significant impact on the fields of reproduction and translational research. The porcine species is a valuable model due to their similarities to humans in reproductive physiology and anatomy, and their use in studying fertility, pregnancy, and embryonic development. In addition to their value in research, the porcine specie is also economically important in the agricultural industry [10,11]. Bovines are essential for the agriculture industry’s profitability, and research on bovine reproduction is crucial for the development of assisted reproductive technologies that have significant implications for both animal and human reproductive health [12]. Canine species are often used as a model for studying human reproductive disorders due to their similar reproductive physiology [13]. Furthermore, they are widely utilized in research concerning artificial insemination and reproductive technology [13]. Canines are also frequently used in studies exploring reproductive aging and the effects of environmental factors on reproductive health [14].

The objective of this study was to develop and characterize the 3D culture technique of YS tissue from three main reproductive domestic species in order to establish an experimental system that could aid in the understanding of YS development pathways and provide potential applications for reproductive technology. By establishing the 3D culture system, we can gain insight into the functional mechanisms that the YS has on health and diseases. This development will be a significant stepping stone towards further exploring the mechanisms of YS tissue and their potential applications in reproductive technology.

## 2. Materials and Methods

### 2.1. Animal Sample Collection

All procedures were conducted in accordance with the research protocol approved by the Ethics Committee of Animal Use of the Faculty of Animal Science and Food Engineering at the University of São Paulo (7278301018). YS tissues were collected from embryos between the 20th and 30th day of pregnancy as YS mesenchymal stem cells exhibit higher proliferation rates during this time frame [15,16,17].

A total of four bovine embryos were obtained from gravid uteri in regional slaughterhouses, with the slaughter of the animals being in accordance with the “Regulation of Industrial and Sanitary Inspection of Products of Animal Origin” (RIISPOA, from its Portuguese acronym). To quantify the embryonic days of the obtained fetuses for inclusion in the study, the scale proposed by Gjesdal (1969) was used [18], which involved measuring cranial length to determine the embryo’s age.

For the canine species, 6 gravid uteri from healthy dams between 3 and 5 years old were collected during castration campaigns in Pirassununga-SP. Throughout the castration process, all animals were administered recommended sedation and anesthesia, and the procedure was conducted using aseptic techniques in accordance with Brazil’s National Council for the Control of Animal Experimentation (CONCEA). To confirm the embryonic day of the collected embryos, the adapted scale proposed by Lopate et al. (2008) was employed, which involved measuring the crown-rump length of the fetus [19]. In this study, YS tissues from each gravid uterus were pooled to represent one biological replicate. With six gravid uteri, a total of six YS tissue biological samples were obtained.

In the case of the porcine species, four gravid uteri were collected from sows that had been previously induced for pregnancy. These animals were part of the extensive pig production program at the FZEA campus, and all handling and reproduction techniques adhered to the guidelines established by CONCEA. Furthermore, the transportation and slaughter of these animals complied with RIISPOA regulations and took place within the FZEA school slaughterhouse. The embryos were collected on the 30th day of pregnancy.

### 2.2. Isolation Protocol and Study Design

The procedure comprised two phases: (1) dissection of embryos and collection of YS tissues, and (2) digestion of YS tissues. First, embryos were dissected along with their respective placental attachments, and the YS was carefully separated from other placental attachments using a scalpel and a light microscope [15,16,17]. Since the YS tissues were collected after the 18th embryonic day, they were all considered secondary YS, composed of endoderm, fetal circulation, and mesothelium. Consequently, the YS explants consisted of cell layers derived from both mesoderm and endoderm (Figure 1).

The isolated YS tissues were then washed with a phosphate-saline solution (PBS) supplemented with 10,000 IU Penicillin/Streptomycin (P/S, Sigma, P4333, St. Louis, MO, USA) and gently macerated to increase the area for enzymatic digestion. The digestion protocol was adjusted for each species. For the canine samples, a tissue/collagenase IV (Sigma, C5138) ratio of 1:1 was used for approximately 1 h and 30 min [15], while for the bovine samples, enzymatic digestion with the same proportion took only 1 h [16]. The porcine samples underwent enzymatic digestion using trypsin 2.5% (Gibco, 15090046, Billings, MT, USA) for 5 min at 37 °C [17]. After the enzymatic process, the YS tissues from each species were centrifuged at 600× *g* for 5 min, and the cell pellet was plated according to three different cell culture methods—a 2D cell culture technique (control technique), a 3D method using Matrigel^®^ (Corning, 356237, Glendale, AZ, USA), and a 3D method with Hanging-Drop (Figure 1).

In this study, we opted to use two distinct 3D in vitro culture techniques based on the nature of the YS tissue. While it develops in a stiff matrix, such as the endometrium, the level of invasion between maternal and fetal tissues varies among species [10,20]. For instance, canine species exhibit endotheliochorial placentation, where the invasion is intermediate. In contrast, bovine and porcine species with epitheliochorial placentation have minimal invasion, with tissues merely in opposition.

We chose Matrigel^®^ as it is a commercially available extracellular matrix (ECM) derived from Engelbreth-Holm-Swarm (EHS) mouse sarcoma [20]. Its complex composition includes laminin, collagen IV, heparan sulfate proteoglycans, and growth factors [21], enabling the creation of a 3D cell culture environment that closely mimics in vivo conditions by supporting cell adhesion, migration, differentiation, and angiogenesis [21]. However, its composition may vary between batches, causing experimental variability, and it is relatively expensive [22]. To supplement Matrigel^®^, we also assessed the Hanging-Drop technique, an alternative method for generating 3D spheroids from cells. In this approach, cells are suspended in a small drop of culture medium and placed on the underside of a Petri dish lid. The drop is suspended by gravity, allowing the cells to aggregate and form a spheroid. This technique offers several advantages, including simplicity, low cost, and the capacity to produce uniform spheroids.

In summary, to establish a 3D in vitro culture of YS tissue from different mammalian species, the YS tissues from canine, porcine, and bovine species were first isolated to obtain the pool of cells present in YS tissue. Afterward, fresh cell explants from each species were cultured using three different techniques—2D cell culture technique (Section 2.5), 3D culture using Matrigel^®^, and 3D culture using the Hanging-Drop technique (Section 2.3). To determine the best conditions for developing tridimensional structures, eight types of media were evaluated (Section 2.4). Once the optimal medium conditions were established, the resulting structures were assessed to analyze morphology and function.

### 2.3. D Cell Culture Establishment—Matrigel^®^ Method and Hanging-Drop Method

In the Matrigel^®^ technique, fresh YS cells were plated in a 1:25 µL ratio of Matrigel^®^ in 48-well plates with 250 µL of growth media. Typically, two weeks after the establishment of the initial culture, the Matrigel^®^ culture could be subcultured at a 1:5 ratio every week (Figure 2) until passage 3.

For the Hanging-Drop technique, cells were cultured in 20 µL drops of growth media on the lid of a 60 mm dish at a concentration of 1:20 µL. After the establishment of the initial culture, subculturing occurred every 3–5 days (Figure 3) at a 1:3 ratio until passage 3. Subsequently, morphological and functional analyses were performed.

### 2.4. D Cell Culture Establishment—Media Used

To determine the most suitable culture medium for each species, we evaluated eight distinct expansion media. The choice of growth factors and supplements for each medium was informed by previous studies on trophoblast organoid development across different species [23,24], and factors present during early gestation and the initial third of gestation in the three target species [24,25,26,27].

For the culture medium, the following growth factors and supplements were used: Recombinant Mouse Epithelial Growth Factor 50 ng/mL (EGF, Peprotech, AF-100-15, Cranbury, NJ, USA), Recombinant Murine Noggin 100 ng/mL (NOG, Peprotech, 250-38), Recombinant Human Rspondin 1 500 ng/mL (RPOS-1, Peprotech, 120-38), Recombinant Human Fibroblast Growth Factor-10 100 ng/mL (FGF-10, Peprotech, 100-26), Recombinant Murine Hepatocyte Growth Factor 50 ng/mL (HGF, Peprotech, 315-23), TGF-β type I receptor (ALK5-TD) inhibitor 500 nM (A83-01, System Biosciences, ZRD-A8-02, Palo Alto, CA, USA), Insulin Growth Factor II (IGF-II, Peprotech,100-12), Leukemia Inhibitory Factor (LIF, Peprotech, AF100-20), Supplement B27 (B27, Gibco, 12587010), N-2 Supplement (N2, Gibco, 17502048), Nicotinamide (Sigma, N0636), Glutamine (Glut, Sigma, G7513), and N-Acetyl-L-cysteine (L-Nac, Sigma, A9165).

In total, we tested eight different growth media with varying addition of specific growth factors (Table 1). Additionally, we used the following supplements in all expansion basal media: B27, N2, Nicotinamide, Glutamine and L-Nac, EGF, NOG, RPOS-1, FGF-10, and HGF, together with the culture media F-12 (Gibco, 21127022).

The optimal medium for each species in 3D culture was chosen based on the following criteria, using cells from the 3D Matrigel^®^ method: (1) medium acidosis due to nutrient consumption, and (2) the number of structures formed during 3D culture across different passages (Figure 4). To perform an objective analysis, we counted organoid-shaped structures in three distinct images of biological replicates for each culture medium and species. These images were captured at 10× magnification and analyzed using ImageJ software [28]. Measurements were taken from each 3D culture technique across three first passages. The medium that supported the development of cultures with the highest number of structures formed during subculturing was selected as the most suitable medium for each species.

Cell cultures were maintained in an incubator at 5% CO_2_ and 37 °C throughout the cultivation period. Both 3D culture techniques for all animal samples were cultured up to passage 3 to determine the most suitable growth media and carry out the subsequent characterization analyses.

### 2.5. D Cell Culture—Control Method of YS Tissue Culture

To establish a control cell culture technique, we chose to use the 2D cell culture model. Cells isolated from freshly collected YS tissue were plated in T-75 flasks with growth media specific to each species. For canine samples, the medium was composed of Alpha-MEM (α-MEM, ThermoFisher, 12571063, Waltham, MA, USA), 15% Fetal Bovine Serum (FBS, ThermoFisher, A5256801), 1% Penicillin/Streptomycin (P/S), 1% Glutamine, and 1% non-essential amino acids solution (NEAA, ThermoFisher, 11140076) [15]. For bovine YS tissue, the medium contained α-MEM, 10% FBS, 1% NEAA, 1% P/S, and 1% BME Amino Acids Solution (Sigma, B6766) [16,17]. Finally, for porcine species, the medium consisted of α-MEM, 15% FBS, 1% P/S, 2% Glutamine, and 25 µg Amphotericin B (AMB, ThermoFisher, 15290018) [17].

The time it took for each culture to reach 80% confluence aligned with previous studies on 2D cultures of YS tissue from canine, bovine, and porcine species [15,16,17]. Cells were subcultured using 1 mL of 0.25% Trypsin solution (25200072, Life Technologies, Carlsbad, CA, USA), incubated for 5 min, and transferred to a 15 mL conical tube for centrifugation at 300× *g* for 5 min. The supernatant was then discarded.

The cell cultures were maintained in an incubator at 5% CO_2_ and 37 °C. All samples were cultured until passage 3 to obtain a more homogeneous in vitro culture, after which characterization analysis was performed.

### 2.6. Morphological Characterization—Optical Microscopy

To track the progress of the cultures over time and compare the three cell culture techniques—2D culture, 3D Matrigel^®^, and 3D Hanging-Drop—we utilized the EVOS M500 microscope. The cultures were assessed weekly to perform media evaluation assays and measure the size of the structures generated by each 3D technique. For comparing structure sizes across different 3D techniques, three distinct images of biological replicates were taken for each culture medium and species. These images were captured at 10× magnification and analyzed using the software, ImageJ [28].

### 2.7. Morphological Characterization—Scanning Electron Microscopy

Scanning electron microscopy (SEM) is a powerful technique for characterizing the morphology and organization of cells in 3D cultures as it enables visualization of surface topography and intricate details of cells and their extracellular matrix. SEM generates high-resolution images of cells in 3D cultures, offering valuable insights into the structure and organization of cells within the culture. SEM was employed to examine and compare the morphology of cells and structures produced by different culture techniques to the natural structure of the YS tissue. In particular, the SEM analysis focused on the organization of various cell types, such as cuboidal and squamous epithelial cells found in different parts of the YS tissue.

To prepare samples for scanning electron microscopy, the samples were initially fixed in 2.5% glutaraldehyde for 4 h at 4 °C. They were then dehydrated using an ethanol gradient with increasing concentrations every 10 min: 30%, 50%, 70%, 90%, 90%, and 100%. After dehydration, the material was stored in a sealed Petri dish until analysis using the TM3000 scanning microscope.

### 2.8. Morphological Characterization—Immunohistochemistry

The immunohistochemistry technique (IHC) was employed to examine the localization of ABCA1 and SLC39A7 proteins, which are the primary transport proteins responsible for transporting cholesterol and minerals, respectively, in the YS membrane [29]. The antibodies used were anti-ABCA1 (ab7360, Abcam) and anti-SLC39A7 (ZIP7, ab117560; Abcam) at a dilution of 1:500 for anti-ABCA1 and 5 μg/mL for anti-SLC39A7, respectively. Histological sections with a thickness of 5 μm were arranged on slides, which were dewaxed in an oven at 37 °C and xylene, followed by hydration in ethyl alcohol and washing with distilled water. The histological sections were subjected to a 0.01 M citric acid solution with a pH of 6.0 and a temperature of 95 °C for 30 min, followed by blocking endogenous peroxidase with 3% hydrogen peroxide for 60 min. After this period, the primary antibody (each at the dilution described above) was incubated for 16 h at 4 °C in a humid chamber. The slides were then washed in buffered saline, and the secondary antibody AlexaFluor^®^488 (ab150077) was added and incubated for 30 min at room temperature. The reaction was revealed with DAB (Chromogen/Substrate Bulk Pack, ScyTek Laboratories, Logan, UT, USA) for 2 min and counterstained with hematoxylin. In the negative control, the primary antibody was omitted, and all slides were counterstained with hematoxylin.

### 2.9. Characterization—Genetic Expression

In this study, the selected genes of interest were Alpha-fetoprotein (*AFP*) and Hemoglobin subunit zeta-like (*HBZ*), which encode proteins produced by the YS with specific functions. *AFP* is involved in metabolic function as it is produced in the YS and later in the embryonic liver [30]. On the other hand, *HBZ* is associated with hematopoietic function, specifically with the formation of the first blood islets [31].

To standardize the gene expression data, housekeeping genes were used for each species. Beta-2-microglobulin (B2M) was used for dogs, while actin beta (ACTB) was used for bovine and porcine species. A complete list of all genes used in the study can be found in Table 2. RNA extraction from the samples was conducted using the RNeasy^®^ Plus kit (Qiagen^®^, 73404, Hilden, Germany), following the manufacturer’s protocol. The RNA concentration was assessed by analyzing the extracted samples on a Nanodrop 1000. (Thermo Fisher Scientific, Waltham, MA, USA), followed by cDNA synthesis. The High-Capacity RNA-to-cDNA Kit (Thermo Fisher Scientific, 4387406) was utilized for cDNA synthesis, adhering to the manufacturer’s recommendations. The RT-qPCR technique was employed to evaluate gene expression. Reactions were carried out on the Step One RT-PCR System (Thermo Fisher Scientific, 12594025), using the QuantiNova SYBR Green PCR Kit (Qiagen, 208052). All samples were analyzed in triplicate. After amplification, the Sequence Detection Software, version 2.0 (Applied Biosystems, Waltham, MA, USA), was utilized for data analysis. The results were obtained as threshold cycle values (Ct), and the expression levels were calculated using the 2-ΔCt method [32].

### 2.10. Statistical Analysis

The gene expression between cell culture techniques was compared using a one-way analysis of variance (two-way ANOVA) followed by Turkey’s multiple comparison test. Statistical significance between groups was considered when *p*-value was less than 0.05, denoted by *, *p* ≤ 0.05; ** for *p* ≤ 0.01; *** for *p* ≤ 0.001, and **** for *p* ≤ 0.0001.

## 3. Results

### 3.1. 3D Model Culture Establishment

To establish the optimal 3D model for YS culture, three different cell culture techniques were employed: (1) the conventional monolayer culture (2D method), (2) the Matrigel^®^ hydrogel-based three-dimensional culture (3D Matrigel^®^), and (3) the Hanging-Drop method-based three-dimensional culture (3D Hanging-Drop). The 2D culture method was included because it is the established method for culturing YS-derived stem cells [15,16,17].

In order to identify the optimal 3D media for each species, the number of formed structures was analyzed in passages 1 through 3 (Table 3) using the Matrigel^®^ Method. Media promoting adequate cell growth were determined for each species. In the 3D cell culture models, medium #2 was the most suitable for canine YS samples, consistently fostering the development of the highest number of 3D structures across all three passages. While media #1 and #3 showed a good number of structures initially, only medium #2 (containing A83-01 and IGF-II) maintained cell growth in subsequent passages. For porcine species, medium #3 (containing A83-01, IGF-II, and LIF) was the best option, enabling sustainable 3D structure growth and development using the Matrigel^®^ method until passage 3. Medium #1, though exhibiting more structures initially, declined in subsequent passages. Finally, for bovine species, medium #4 (containing only A83-01) was the optimal choice, allowing the development and maintenance of structure numbers across various passages (Table 3).

Using the chosen model, the primary culture duration for the two proposed 3D culture models was 18 days for canine species, 12 days for porcine species, and 16 days for bovine species with the 3D Matrigel technique, respectively. In contrast, with the Hanging-Drop technique, the primary culture duration was approximately 7 days for canine species, 6 days for porcine species, and 9 days for bovine species, respectively. The subsequent subculture passages were performed every week for organoid culture and every 3–5 days using the Hanging-Drop technique.

For the 3D culture of YS tissue cells, all species examined in this study required a medium supplemented with A83-01. Additionally, only canine and porcine species needed IGF-II supplementation, and solely porcine species necessitated LIF supplementation (Table 4). The dependence of all three species on A83-01 is consistent with previous 3D cell culture studies, as A83-01 is essential for supplementing the 3D culture of various tissues, such as the endometrium [33,34], trophoblast [23,24], and placenta [34]. Additionally, IGF-II supplementation was found to be crucial only for canine and porcine species. IGF-II is a growth-promoting hormone during mammalian pregnancy that aids in mesothelium formation during embryo development and performs a significant role in canine pregnancy [26] and porcine embryological development [35]. Lastly, only porcine species required LIF supplementation, which aligns with previous in vitro studies demonstrating that LIF is expressed by the porcine trophectoderm to assist in the adherence and viability of the embryo [36].

### 3.2. Morphology and Cellular Conformation

Using light microscopy, we observed that YS tissue stem cells cultured in 3D techniques resulted in cystic, balloon-like morphologies in three-dimensional extensions (Figure 5). Furthermore, after measuring the size of the structures using Image J software, we discovered that canine species exhibited structures with a size of 38,273.87 µm ± 2586.89 µm in the 3D Matrigel^®^ technique, while in the 3D Hanging-Drop technique, these structures were approximately half the size, measuring 19,821.13 µm ± 1395.23 µm. For porcine species, the achieved sizes were 33,654.58 µm ± 3875.09 µm, and 14,040.99 µm ± 1608.67 µm for Matrigel^®^ and Hanging-Drop techniques, respectively. For bovine species, the sizes were similar with 22,911.84 µm ± 3409.24 µm, and 11,382.27 µm ± 1020.91 µm for Matrigel^®^ and Hanging-Drop techniques, respectively (Figure 5).

Additionally, SEM was employed to compare the organizational structure of the 3D in vitro techniques. It was observed that both 3D cultures generated a cell conformation similar to that found in YS tissue (Figure 6). For canine species, both 3D techniques resulted in a highly digitized external conformation, featuring microvilli and elongated cells, and short branching processes (Figure 6A), characteristic of the mesothelium layer of the YS [20]. The SEM analysis of porcine YS tissue revealed a less digitized and more elongated appearance, resembling the mesothelium layer of the YS tissue in this species [37]. Moreover, both 3D techniques derived from porcine YS displayed a conformation more akin to the mesothelium layer (Figure 6C), with more pronounced flatter poles and sparse, irregular distribution of microvilli [37]. In bovines, the YS tissue exhibited more rounded and fused microvilli, and a cubic shape resembling the endoderm layer of the YS (Figure 6B), consistent with a previous report by Galdos-Riveros et al. in 2012 [38]. Furthermore, both 3D culture methods of bovine species produced structures with an external composition resembling the mesothelium layer, which contrasted with the results reported by Mançanares et al. in 2019 [39], where the 3D culture of bovine YS generated structures resembling blood vessels. 

### 3.3. Characterization—Transport Markers

The selection of lipid transporter (ABCA-1) and mineral transporter (SLC39A7) was based on a recent study that identified the presence of transporters responsible for the transport of various substances, such as amino acids, cholesterol, metals, and anions, in human, mouse, and chick YSs. These transporters were found to be conserved across species [29] and are associated with drug resistance and absorption [40].

The expression of proteins ABCA-1 and SLC39A7 was observed in all in vitro models of different species (Figure 7). In YS tissues, the endoderm layer exhibited the primary expression of these markers, while in 3D in vitro methods, their expression was higher in the outer layer of Matrigel^®^-derived structures and randomly distributed in Hanging-Drop formations. In the 2D culture, the SLC39A7 marker was distributed across the cell surface, while the ABCA-1 marker was more concentrated around the cell nucleus (Figure 7). However, the multicellular arrangement of markers observed in the original tissue and 3D methods was not seen in the 2D culture. The results indicate that the expression pattern of these markers is more complex in 3D cultures than in conventional 2D models. Therefore, the 3D in vitro models of YS display a higher degree of cell differentiation than those obtained by 2D cultures.

### 3.4. Characterization—Genetic Expression

For both canine and porcine species, in general, the 3D Matrigel^®^ technique provided a higher *HBZ* expression, and this may be attributed to the fact that *HBZ* is mainly expressed in the blood islands of the YS [5,41], which have similar characteristics of higher stiffness and tension to the 3D Matrigel^®^ method. In contrast to the findings in other species, in bovine samples, the expression of both *AFP* (Figure 8B) and *HBZ* (Figure 8E) was significantly higher in the 2D cell culture compared to the 3D models studied (*p* < 0.05). This observation was different from the pattern reported by Mançanares et al. (2019) [39] and Potapova et al. (2007) [41], where an increase in gene expression was observed in 3D cultures. However, none of the cell culture methods were able to replicate the same *AFP* expression pattern found in the YS tissue (Figure 8A,C,E).

## 4. Discussion

The YS tissue is a crucial extraembryonic membrane that supports hematopoiesis and provides nutrients in early embryonic development. Culturing YS tissue in 3D offers numerous advantages, as it contains diverse cell populations, including hematopoietic, endothelial, and mesenchymal cells [5]. Thus, 3D culture enables the maintenance of cellular heterogeneity and the study of cell–cell interactions. The YS has been linked to early miscarriage [3] and studying the YS tissue through 3D in vitro culture can allow investigation into disease mechanisms and the development of novel therapies. Additionally, the YS is important in transporting nutrients and waste products, making it a valuable target for toxicity testing [9]. Developing a 3D in vitro culture of YS facilitates toxicity assessments and the development of safer drugs.

In this study, we established a reproducible protocol for generating 3D cultures of YS tissue using specific culture conditions and media. The light microscope morphology and immunohistology cuts presented by the 3D Matrigel^®^ method are consistent with previous studies that reported the generation of organoids from trophoblast and endometrium, displaying a spherical shape with distinct layers arranged around a central cavity [23,34]. The Hanging-Drop technique originated structures with a shape and morphology similar to those described by Foty et al. 2011 [42], which derived spheroids from tumor cells. These structures are composed of aggregated cells that form a three-dimensional structure with a central core and outer layer of cells, similar to those visualized in our study.

Regarding the size of the formed structures, the Matrigel^®^ method produces structures of a size of 31,613.43 µm ± 7881.79 µm, while the Hanging-Drop method generates structures half the size of the Matrigel^®^ method^®^—15,081.46 µm ± 4314.57 µm. These sizes are consistent with organoid and spheroid descriptions, respectively [23,33,34,42,43]. The Matrigel^®^-derived 3D method forms more complex and organized structures that resemble the description and classification of an organoid model, while Hanging-Drop structures resemble spheroid descriptions. The time to obtain the formed structures varied between the two techniques. The Matrigel^®^ method takes around 15 days to establish the first culture, and subsequent subculturing takes approximately one week, which is consistent with previous reports [23,33]. The Hanging-Drop method has a faster establishment of the first culture at around seven days, and passages were performed every five days, also consistent with previous reports [22].

Both transport markers used in this study were found in both the endoderm and mesothelial layers of YS tissue, so using only IHC assay would not be possible to differentiate between the endoderm and mesothelium layer [29]. However, SEM analysis indicated that both 3D methods originate structures that assemble from the mesothelium layer of YS tissue. The low expression of *AFP* indicates a lack of endoderm differentiation as *AFP* is specifically expressed in endodermal cells of the YS during early embryonic development [44].

Based on the results, the structures derived by the Matrigel^®^ method would be suitable for studying YS transport elucidation mechanisms and disorders due to their better organizational conformation. Meanwhile, the Hanging-Drop technique, due to its better capacity to develop spheroids and its lower cost, can provide a platform for drug screening and toxicity testing, offering a more physiologically relevant environment than 2D cell cultures. To improve the methods described, it would be beneficial to further confirm the obtained layers by using markers, such as Wilms tumor protein 1 (WT1), for mesothelium differentiation [45] and SOX17 for endodermal differentiation [46]. Additionally, for the Matrigel^®^ method, assessing genes associated with stemness or self-renewal, such as SOX2, NANOG, or OCT4 [47], which were not assessed in the present research, would be beneficial.

In conclusion, the establishment of a reproducible protocol for generating 3D cultures of YS tissue using specific culture conditions and media offers numerous advantages in studying YS tissue. Both the Matrigel^®^ and Hanging-Drop methods can generate structures that resemble organoids or spheroids, respectively. The Matrigel^®^ method generates more complex and organized structures suitable for studying transport elucidation mechanisms and disorders, while the Hanging-Drop technique is suitable for drug screening and toxicity testing. Further studies on markers for differentiation and genes associated with stemness and self-renewal will improve the methods and help in the understanding of YS tissue development and disorders.

## 5. Conclusions

Our study evaluated three in vitro methods for culturing YS tissue, and discovered that the 3D technique model demonstrated superior cell organization and morphological similarity to the original tissue compared to the standard 2D model. The preliminary findings from this research seek to establish a methodology for cultivating 3D structures from mammalian YS tissue, offering an initial characterization of the resulting structures, and outlining potential applications and future directions.

## Figures and Tables

**Figure 1 animals-13-01435-f001:**
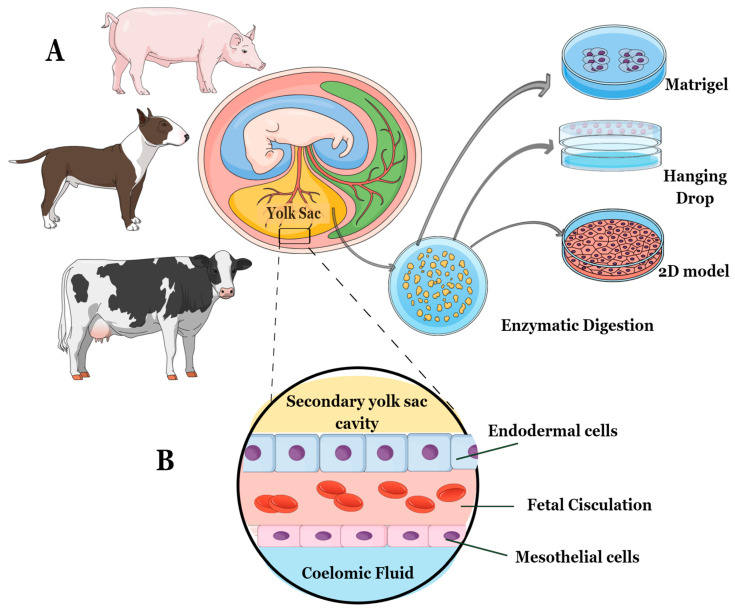
The study followed this experimental design: (**A**) Embryos between 20 and 30 gestational days were selected for the study. For the canine and porcine species, embryos from the same gravid uteri were combined to minimize the impact of individual differences. (**B**) The YS tissue used in the study was obtained from specific cell layers composed of both mesoderm and endoderm. These cell layers included endoderm, fetal circulation, and mesothelium. After isolating fresh YS tissue from each species, the explants were cultured using three different methods—2D technique, 3D Matrigel^®^, and 3D Hanging-Drop technique.

**Figure 2 animals-13-01435-f002:**
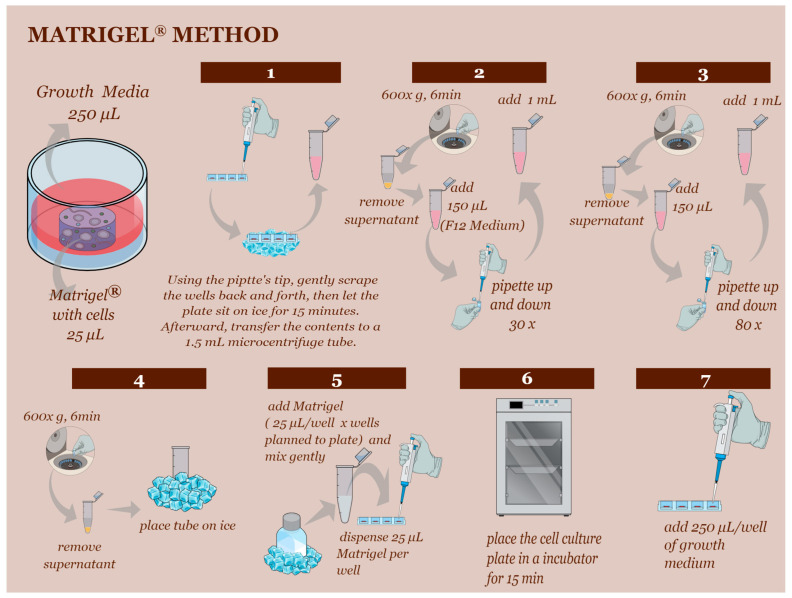
The subculturing technique using Matrigel^®^ technique. (Step 1) Begin by scraping the Matrigel^®^ and media to separate and homogenize them. Place the plates on ice for 15 min to liquefy the Matrigel^®^. (Step 2) Transfer the liquefied Matrigel^®^ to a 1.5 mL centrifuge tube and centrifuge at 600× *g* for 6 min. Add 150 µL of F12 basal medium and follow the instructions in steps 2, 3, and 4. (Step 4) Before proceeding to step 5, ensure the Matrigel^®^ has been on ice for at least 30 min to bring the reagent to a liquid state. (Step 5) Use a 1:5 proportion to subculture the cells. Continue with steps 6 and 7 as described.

**Figure 3 animals-13-01435-f003:**
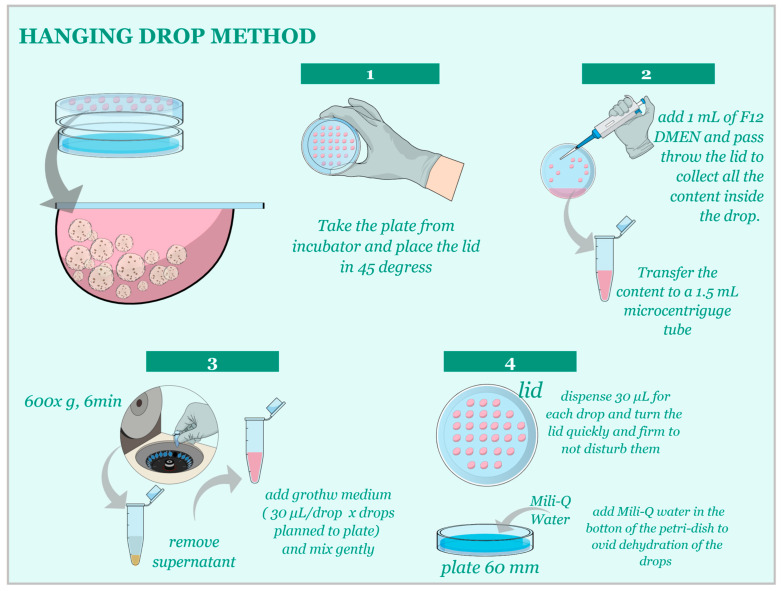
(Step 1) Carefully observe the spheroids formed in the hanging drops to determine when they are ready for subculturing. This typically occurs when the spheroids reach an optimal size and cell density. (Step 2) Prepare a collection dish by adding an appropriate volume of fresh growth media to a 60 mm dish. Carefully remove the lid with hanging drops from the original dish, taking care not to disturb the drops. Holding the lid over the collection dish, gently wash the hanging drops off the lid by pipetting a small volume of the media from the collection dish onto each hanging drop, allowing the spheroids to fall into the media below. (Step 3) Once all spheroids have been collected in the media, centrifuge for 600× *g* for 6 min, remove the supernatant, and dilute the collected spheroids in the appropriate volume of fresh growth media to achieve a 1:3 proportion for subculturing. (Step 4) Prepare new hanging drops with the diluted spheroids on the lid of a new 60 mm dish. Monitor the spheroids’ growth and repeat the subculturing process as needed.

**Figure 4 animals-13-01435-f004:**
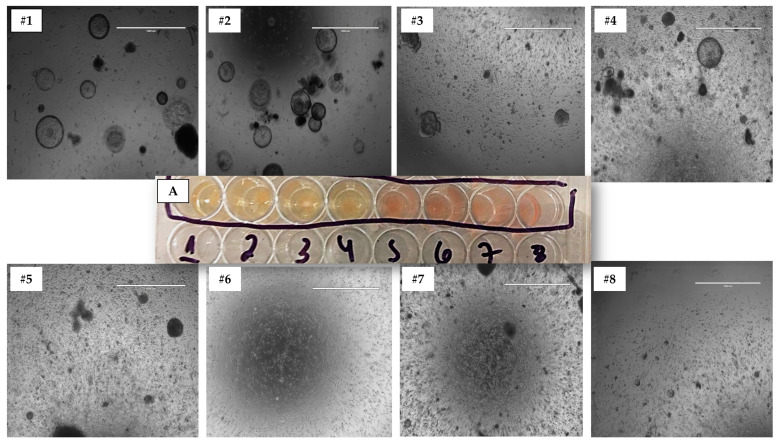
This figure illustrates the process of determining the optimal medium for 3D cultures in a rep-presentative study. It demonstrates the assessment of (**A**) medium acidosis, and the number of structures formed across eight different types of media (**#1**, **#2**, **#3**, **#4**, **#5**, **#6**, **#7**, and **#8**) using the Matrigel^®^ technique. The image shows one of the technical replicates from a canine sample in passage 3. All species and media images were evaluated in technical triplicates for each biological sample at 10× magnification.

**Figure 5 animals-13-01435-f005:**
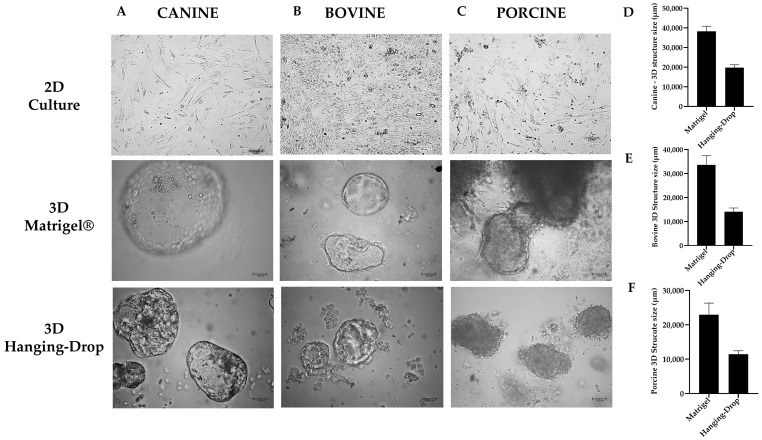
Representative images of each studied group were taken at 10× magnification. These images display the morphology of different methods used in this study for (**A**) canine species, (**B**) bovine species, and (**C**) porcine species. Additionally, the size of the structures formed using the 3D culture techniques were compared for (**D**) canine species, (**E**) bovine species, and (**F**) porcine species. Each image represents one of the replicates for each species under different culture conditions (3D Matrigel^®^, 3D Hanging-Drop, and 2D culture technique) at passage 3.

**Figure 6 animals-13-01435-f006:**
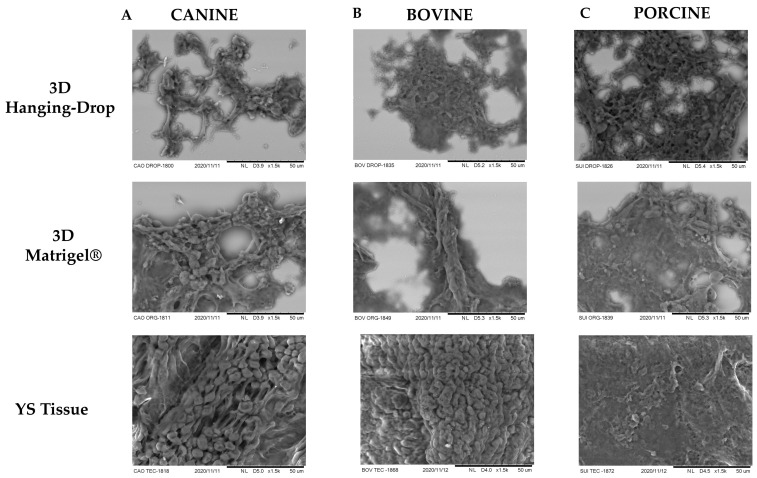
Scanning electron microscopy images were taken at 50× magnification for different species and culture techniques. Each image represents one of the replicates for each species under various culture conditions in passage 3: (**A**) canine species, (**B**) bovine species, and (**C**) porcine species. The total number of samples for each species and technique was *n* = 3. To compare the conformational organization of different formed structures, YS tissue from each species was used as a reference. The SEM analysis of YS tissue from canine and porcine species resembled a mesothelium layer, whereas the bovine YS sample resembled an endothelium layer of YS. Both 3D culture techniques resulted in structures exhibiting conformational characteristics of the mesothelium layer, with a less digitized and more elongated appearance.

**Figure 7 animals-13-01435-f007:**
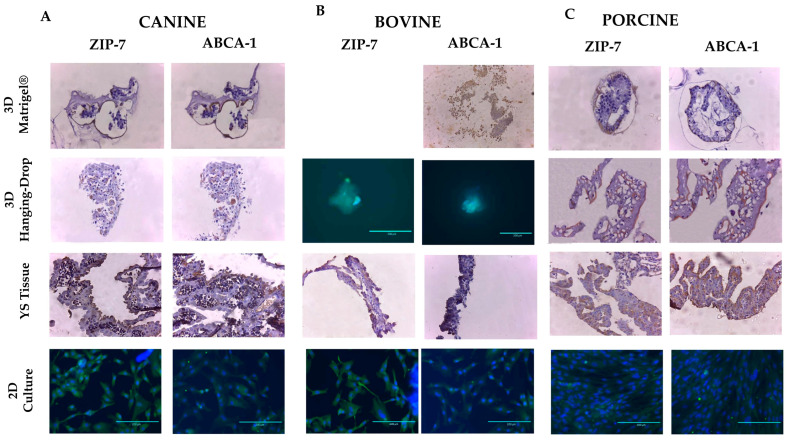
The expression of membrane markers ZIP-7 (SLC39A7 protein) and ABCA-1 protein was analyzed by immunohistochemistry and immunocytochemistry in each studied group, including (**A**) canine, (**B**) bovine, and (**C**) porcine species. Immunohistochemistry images were captured at 10× magnification, while immunocytochemistry images were captured at 40× magnification (*n* = 3).

**Figure 8 animals-13-01435-f008:**
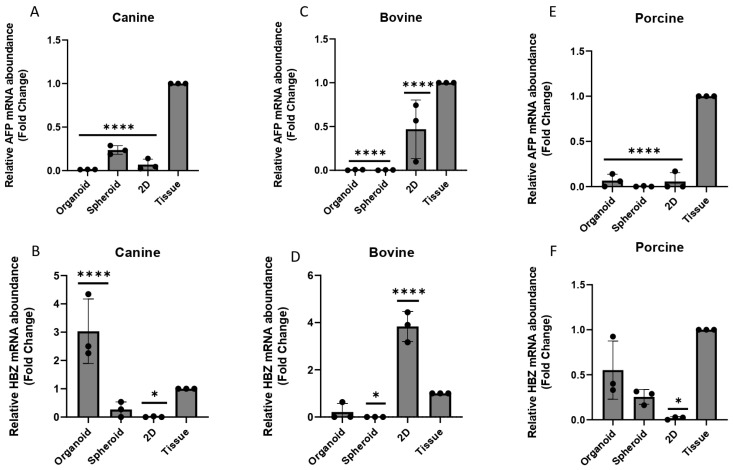
The relative expression of *AFP* and *HBZ* in different species and cell culture techniques (*n* = 3 for each species and in vitro technique analyzed). (**A**,**D**) canine species, (**B**,**E**) bovine species and (**C**,**F**) porcine species. (*) represents statistical differences *p* < 0.05, (****) represents statistical differences *p* < 0.0001 in relation to the YS tissue.

**Table 1 animals-13-01435-t001:** Reagents and concentrations used in the study.

Compound	Concentration	Vendor	Cat #
Supplement minus vitamin A (B27)	1%	Gibco	12587010
N2	1%	Gibco	17502048
Nicotinamide	10 nM	Sigma	N0636
Glutamine	1%	Sigma	G7513
L-Nac	1%	Sigma	A9165
F-12 culture medium	-	Gibco	21127022
Recombinant Human Noggin (NOG)	100 ng/mL	Peprotech	120-10C
Recombinant Human R-Spondin-1 (RPOS-1)	500 ng/mL	Peprotech	120-38
Recombinant Human FGF-10 (FGF-10)	100 ng/mL	Peprotech	100-26
Recombinant Murine HGF (HGF)	50 ng/mL	Peprotech	315-23
Recombinant Murine EGF (EGF)	50 ng/mL	Peprotech	315-09
ALK-4, -5, -7 inhibitor (A83-01)	500 nM	Sigma	SML0788
Recombinant Human IGF-II (IGF-2)	50 ng/mL	Peprotech	100-12
Recombinant Human LIF (LIF)	50 ng/mL	Peprotech	300-05
Matrigel^®^	25 µL/well	Corning	356231

N2 (N-2 Supplement), L-Nac (N-Acetyl-L-cysteine), NOG (Recombinant Murine Noggin), RPOS-1 (Recombinant Human Rspondin 1), FGF-10 (Recombinant Human Fibroblast Growth Factor-10), HGF (Recombinant Murine Hepatocyte Growth Factor), EGF (Recombinant Murine Epithelial Growth Factor), A83-01(TGF-β type I receptor (ALK5-TD) inhibitor), IGF-II (Insulin Growth Factor II), LIF (Leukemia Inhibitory Factor).

**Table 2 animals-13-01435-t002:** Sequence of primers used in the study.

Genes	ID Access		Sequence
Canis lupus familiaris			
Alpha-fetoprotein (*AFP*)	NM_001003027.1	forward	TTCCAAGTTGCAGAACCCGT
		reverse	CCATAGTGGGCAGCCAAAGA
hemoglobin subunit zeta-like (*HBZ*)	XM_003639130.3	forward	TCCCACTCAGCTCCACCAT
		reverse	ATCTTGCCCCACATGGACAG
beta-2-microglobulin (*B2M*)	NM_001284479.1	forward	CTGGCGACGGCTGGTTT
		reverse	TCTGCTGGGTGTCGTGAGTA
Bos taurus			
Alpha-fetoprotein (*AFP*)	NM_001034262.2	forward	GGGAAAATTTGGACCCCGGA
		reverse	CCAGCACGTTTCCTTTGCAG
hemoglobin subunit zeta (*HBZ*)	XM_002683810.5	forward	AAGTTCCTGTCTCACTGCCTG
		reverse	GACGCCGGATACAATCGACA
actin beta (*ACTB*)	XM_005225005.1	forward	CTTCCTGGGTGATCTGCCTT
		reverse	CCGTGTTGGCGTAGAGGTC
Sus scrofa			
Alpha-fetoprotein (*AFP*)	NM_214317.1	forward	AGAGGAAACGTGCTGGAGTG
		reverse	TCAAGTGTGGTGGGCAACTT
hemoglobin subunit zeta (*HBZ*)	XM_003481082.4	forward	ATATAAGGGGACCACGGGGG
		reverse	AATTGTCCTCTCGGCCTTGG
actin beta (*ACTB*)	XM_021086047.1	forward	TGTGGATCAGCAAGCAGGAG
		reverse	CTGCAGGTCCCGAGAGAATG

**Table 3 animals-13-01435-t003:** Number of structures formed across different passages using Matrigel^®^ method.

		Media #							
Specie	Passage #	#1	#2	#3	#4	#5	#6	#7	#8
Canine	Passage 1	11.0 ± 1.2	12.0 ± 1.1	10.0 ± 1.1	8.5 ± 1.1	0.0 ± 0.0	2.8 ± 1.1	0.8 ± 0.7	2.0 ± 1.1
	Passage 2	8.6 ± 1.7	11.2 ± 2.3	7.5 ± 1.6	7.2 ± 1.7	0.0 ± 0.0	2.0 ± 1.3	0.4 ± 0.5	2.2 ± 1.9
	Passage 3	7.7 ± 1.5	11.8 ± 1.5	7.4 ± 2.1	6.7 ± 1.9	0.0 ± 0.0	1.3 ± 0.8	0.3 ± 0.7	0.0 ± 1.0
Bovine	Passage 1	2.5 ± 1.2	1.0 ± 0.8	1.8 ± 1.4	9.7 ± 1.7	0.0 ± 0.0	5.8 ± 0.9	10.2 ± 1.0	1.5 ± 1.0
	Passage 2	1.6 ± 0.7	0.5 ± 0.7	1.5 ± 1.2	9.8 ± 1.4	0.0 ± 0.0	4.0 ± 1.5	9.1 ± 1.0	1.0 ± 0.5
	Passage 3	0.5 ± 0.7	0.3 ± 0.5	1.1 ± 0.9	10.1 ± 1.1	0.0 ± 0.0	3.2 ± 1.8	5.4 ± 1.5	0.0 ± 0.5
Porcine	Passage 1	10.0 ± 1.2	2.8 ± 1.1	13.4 ± 1.4	3.4 ± 1.1	0.0 ± 0.0	0.0 ± 0.0	0.0 ± 0.0	1.4 ± 1.0
	Passage 2	8.1 ± 1.1	2.2 ± 0.6	13.5 ± 1.5	3.2 ± 1.5	0.0 ± 0.0	0.0 ± 0.0	0.0 ± 0.0	1.1 ± 0.6
	Passage 3	7.4 ± 1.0	2.1 ± 1.7	13.5 ± 1.2	2.5 ± 1.0	0.0 ± 0.0	0.0 ± 0.0	0.0 ± 0.0	0.8 ± 0.7

**Table 4 animals-13-01435-t004:** Growth medium used for each species and cell culture technique.

Specie	3D Media	2D Media
Canine	#2—Expansion Media + A83-01 + IGF-2	α-Men + 15% FBS + 1% P/S + 1% glut + 1% AANE
Bovine	#4—Expansion Media + A83-01	α-Men + 10% FBS + 1% P/S + 1% AANE + 1% EAA + 1% BME
Porcine	#3—Expansion Media + A83-01 + LIF	α-Men + 15% FBS + 1% P/S + 2% glut + 25 µg AMB

## Data Availability

The data presented in this study are available on request from the corresponding author.

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
