# Peer review of "Initial Characterization of 3D Culture of Yolk Sac Tissue"

_animals, 2023, doi:10.3390/ani13091435_

Round 1

Reviewer 1 Report (Previous Reviewer 2)

The current version of the manuscript is a much-improved version of the previous one. The message is more clear and more organized. My only concern is the significance and novelty of this study. Since it is an initial study, it will be difficult for readers to replicate this work in their lab. 

Author Response

The current study represents preliminary research focused on developing various methods for the three-dimensional culturing of yolk sac tissue from diverse mammalian species. The yolk sac is a crucial tissue for better understanding histotrophic nutrition, which occurs during the first trimester of mammalian pregnancies – a period characterized by the highest incidence of embryonic loss. By devising new approaches to study and analyze the function of yolk sac tissue, we can pave the way for additional studies and insights into the structure and function of this vital component.

While we acknowledge the preliminary nature of our study, our research group has successfully standardized and replicated the technique described in the manuscript for three distinct species. We are confident that the recent revisions and updates to the methodology have enhanced its clarity, thereby facilitating the reproducibility of our work. Furthermore, in lines 538-543, we briefly comment on potential applications for each 3D cell culture developed. In lines 543-547, we discuss the limitations of our study, enabling future research to build upon and refine our techniques.

Reviewer 2 Report (Previous Reviewer 1)

The authors have addressed many of the previous review concerns that has assisted in justifying and clarifying the experiment and experimental model. This was very helpful in assessing the results and comprehending the discussion.  Issues with resolution were addressed in the Figures, which where easier to read; however, most of the figures were cut off in the resubmitted manuscript. There were serious grammatical structure issues that were not addressed (see below) from the previous review that must be addressed and a few other minor edits.

1.     Fig 2, 3, 6, 7:  figures are cut off in the resubmission

2.     Revised version does not correct grammar issues: Previous review gave examples.  It did not give every single grammar problem that exists in the text. 

a.     Inappropriate use of paragraphs throughout the manuscript, especially in the materials and methods and discussion sections.  One to two sentence paragraphs are poor English grammar and demonstrate a lack of knowledge in the use of paragraphs in the English language. Every new paragraph is supposed to be the ending of one subject and a beginning of another, which is not the use in the manuscript text.  Must be revised.  As is written most paragraphs are an interruption in reading with a continuation of the same subject.  The flow of reading is constantly interrupted due to inappropriate grammatical structure.

b.     Period missing at the end of paragraphs or sentences.

c.     Inconsistent use of Yolk sac and acronym YS

d.     Incomplete parentheses used, i.e., have a beginning OR ending parentheses but missing the other.

e.     Discussion is double spaced, which is inconsistent with the single spacing in the rest of the text.

3.     Line 49: Cell are not ‘stretched’ out in 2D culture systems.  They are spread out over a flat surface which does not simulate the 3D structure of tissue.

4.     Line 99-100: what does ‘and downcast’ mean?

5.     Line 197: ‘…..it was used the stablished described….’ makes no sense. Revise.

6.     Line 296-297: IGF-II can be bound to IGFBP3.  What would it ‘act on’ with a binding protein? 

Author Response

The authors have addressed many of the previous review concerns that have assisted in justifying and clarifying the experiment and experimental model. This was very helpful in assessing the results and comprehending the discussion.  Issues with resolution were addressed in the Figures, which where easier to read; however, most of the figures were cut off in the resubmitted manuscript. There were serious grammatical structure issues that were not addressed (see below) from the previous review that must be addressed and a few other minor edits.

  1. Fig 2, 3, 6, 7:  figures are cut off in the resubmission

Thank you for pointing out the issue with the figures. Indeed, they were not formatted correctly, causing them to be shifted to the left when inserted into the manuscript. We have now resolved this problem, ensuring that the figures are properly aligned and presented.

  1. Revised version does not correct grammar issues: Previous review gave examples.  It did not give every single grammar problem that exists in the text. 
  2. Inappropriate use of paragraphs throughout the manuscript, especially in the materials and methods and discussion sections.  One to two sentence paragraphs are poor English grammar and demonstrate a lack of knowledge in the use of paragraphs in the English language. Every new paragraph is supposed to be the ending of one subject and a beginning of another, which is not the use in the manuscript text.  Must be revised.  As is written most paragraphs are an interruption in reading with a continuation of the same subject.  The flow of reading is constantly interrupted due to inappropriate grammatical structure.

Thank you for your response and understanding. We have adjusted the manuscript, accordingly, ensuring that it is both grammatically correct and effectively presented.

  1. Period missing at the end of paragraphs or sentences.
  2. Inconsistent use of Yolk sac and acronym YS
  3. Incomplete parentheses used, i.e., have a beginning OR ending parentheses but missing the other.

The entire manuscript has been thoroughly reviewed, and all issues have been addressed and resolved.

  1. Discussion is double spaced, which is inconsistent with the single spacing in the rest of the text.

The issue has been addressed: the discussion section has been updated, as indicated on line 499.

  1. Line 49: Cell are not ‘stretched’ out in 2D culture systems.  They are spread out over a flat surface which does not simulate the 3D structure of tissue.

The issue has been addressed, as indicated on lines 51- 53: “In 2D systems, cells tend to be spread out, which does not simulate the tridimensional structure of tissue, impacting many processes, including proliferation, differentiation, and cell death”.

  1. Line 99-100: what does ‘and downcast’ mean?

The appropriate term for the methodology is "slaughter." The entire section 2.1 (lines 88-114) has been thoroughly reviewed and revised accordingly.

  1. Line 197: ‘…..it was used the stablished described….’ makes no sense. Revise.

Thank you for the correction. The description of the 2D cell culture method has been modified and included in a new section, "2.5. 2D Cell Culture – Control Method of YS Tissue Culture," spanning lines 266-283. The previously mentioned issue has been successfully addressed.

  1. Line 296-297: IGF-II can be bound to IGFBP3.  What would it ‘act on’ with a binding protein? 

Due to the modifications of the manuscript, the previous statement was removed from the manuscript. The new description about IGF-II supplementation is now described on lines 398 – 402. New text reads, “Also, IGF-II supplementation was found to be crucial only for canine and porcine species. IGF-II is a growth-promoting hormone during mammalian pregnancy that aids in mesothelium formation during embryo development and plays a significant role in canine pregnancy [26] and porcine embryological development [35]” “Also, IGF-II supplementation was found to be crucial only for canine and porcine species. IGF-II is a growth-promoting hormone during mammalian pregnancy that aids in mesothelium formation during embryo development and plays a significant role in canine pregnancy [26] and porcine embryological development [35]”

Reviewer 3 Report (New Reviewer)

 The study describes and compares different methods to establish yolk sac-like in vitro cultures. The study itself is very promising and also comprehensive and could contribute significantly in 3D cell culture technology.

However, there are some serious shortcomings. Some of the figures are not fully illustrated, so that the paper cannot be fully reviewed. In addition, some methods need to be described in more detail. The discussion should be completely revised as it only provides a description but no discussion of the actual results.

Comments in Detail:

Abstract

 Writing out abbreviations in the abstract, such as SLC and ABCA-1, SLC and ABCA-1, AFP.

Please include more keywords

Introduction

Why some areas/sections are marked in yellow throughout the manuscript?

Methods:

Although it is "only" the killing of animals, the euthanasia and killing method should be described in more detail for ethical reasons. How were the animals anesthetized and killed. Did the 6 dog embryos (line 102) come from one dam or 6? How old were the dams? Where did the ethical approval number come from - university - state agency? What animal welfare law was followed? Survival rate of female dogs after castration.

Fig2 is not completely visible – step 3 is missing

Fig 3 is not completely visible

Line 126: what does mean “according to the culture type”? Was each cell culture type done with the Matrigel method, Hanging drop etc? Also in section 2.3 is not mentioned, if cells of all three animal species were used for both methods

Line 206: how long did each passage take? Were the subsequent experiments with the different cultivation experiments only performed in passage 3? I.e. up to P3 the cells were kept in monolayer and only transferred to 3D from P3 onwards?

Line 210: three techniques – you mean 2D, matrigel and hanging drop – please add this information like provided in line 278ff

 Results:

283ff is already the final result – is this based on all examinations?

Lines 287-302 belong to the introduction or discussion. In the results section, there should be a description of your observation. And not just mention the final statement.

What was the growth of your 3D tissues like in the experimental series - was there a significant difference between the different media, etc.? Provide a table and pictures of the results you obtained from the experiments described in section 2.3 and 2.4, 2.5.

Figure 4 is not from each group studied, but only from one trial of each animal. What is the difference between spheroids and organoids? When did you obtain them? Were they part of the same experimental trial? Or were the organoids obtained from matrigel and the spheroids from hanging drop? Please provide more information in the text and figures to understand the method and your final statement.

Fig 5: could you provide more description of the pictures within the pictures? the pictures are also very blurry, so that the structures are not clearly visible

Fig 6 not fully visible - the descriptions in the text can therefore not be understood or reviewed

Fig 7 not fully visible - the descriptions in the text can therefore not be understood or reviewed

Discussion:

In the introduction, it was mentioned that they are growing these cell cultures to understand miscarriages. Now it seems to be because of diabetic embryopathy and congenital heart disease, tox. testing. Of course there can be more than one target, but within one publication you should stick to one topic.

Moreover, the further explanation is not a real discussion. It is only a description of the 3D method. Some sentences would rather fit into the method section to provide more detailed information about the manufacturing process.

Furthermore, the authors do not elaborate on their results, but rather state what should have been done further in order to make valid statements.

In my opinion, the discussion should be fundamentally revised, as it does not provide a discussion of the existing results.

Author Response

The study describes and compares different methods to establish yolk sac-like in vitro cultures. The study itself is very promising and also comprehensive and could contribute significantly in 3D cell culture technology.

However, there are some serious shortcomings. Some of the figures are not fully illustrated, so that the paper cannot be fully reviewed. In addition, some methods need to be described in more detail. The discussion should be completely revised as it only provides a description but no discussion of the actual results.

Thank you so much for all the points and corrections. Your comments on the text have truly helped us improve the description and comprehension of the study. We have addressed all the points raised.

Regarding the figures, indeed, they were not formatted correctly when we added them to the manuscript, causing them to shift to the left. This issue has been resolved.

In terms of methodology, we have reorganized and expanded some sections to enhance the description and explanation of the techniques, as well as to provide additional details and figures that elucidate the protocol used. In the revised version of the manuscript, the methodology section is divided into the following subsections:

2.1. Animal Sample Collection – Line 88

2.2. Isolation Protocol and Study Design – Line 117

2.3. 3D Cell Culture Establishment – Matrigel® Method and Hanging-Drop Method – Line 174

2.4. 3D Cell Culture Establishment – Media Used – Line 209

2.5. 2D Cell Culture – control method of YS tissue Culture – Line 266

2.6. Morphological Characterization – Optical Microscopy – Line 287

2.7. Morphological Characterization – Scanning Electron - Line 296

2.8. Morphological Characterization – Immunohistochemistry – Line 312

2.9 Characterization – Genetic Expression – Line 331

2.10  Statistical Analysis – Line 356

As for the discussion section (line 500), it has been thoroughly revised. Although our study is innovative and we cannot directly compare our results with previous findings in the field, we have discussed the outcomes in relation to similar reproductive studies that involve Matrigel and Hanging-Drop techniques within the fields of reproduction and biomedicine.

Comments in Detail:

Abstract

Writing out abbreviations in the abstract, such as SLC and ABCA-1, SLC and ABCA-1, AFP.

The issue has been addressed. The revised version of the manuscript states, “Regarding the IHC assay, all in vitro models were able to express zinc and cholesterol transport markers, although only 3D culture techniques were able to generate structures with different markers pattern, indicating a cell differentiation process when compared to 2D cultures”. This can be found on lines 26-29 of the current manuscript version.

Please include more keywords 

The keywords included: Matrigel, Hanging-Drop, canine, porcine, bovine, stem cell.

Introduction

Why some areas/sections are marked in yellow throughout the manuscript?

In the previous version of the manuscript, the yellow areas indicated the additional sections. However, for this new version, we have extensively revised and modified the manuscript without including any highlighted areas. This decision was made to prioritize better comprehension of the text.

Methods:

Although it is "only" the killing of animals, the euthanasia and killing method should be described in more detail for ethical reasons. How were the animals anesthetized and killed. Did the 6 dog embryos (line 102) come from one dam or 6? How old were the dams? Where did the ethical approval number come from - university - state agency? What animal welfare law was followed? Survival rate of female dogs after castration.

 The issue has been addressed in the new section "2.1. Animal Sample Collection" (lines 88-114).

Fig 2 is not completely visible – step 3 is missing

Fig 3 is not completely visible

 The issue was addressed.

Line 126: what does mean “according to the culture type”? Was each cell culture type done with the Matrigel method, Hanging drop etc? Also in section 2.3 is not mentioned, if cells of all three animal species were used for both methods

The passage has been reformulated, and the new phrase can be found on lines 131-134: “After the enzymatic process, the YS tissues from each species were centrifuged at 600g for 5 minutes, and the cell pellet was plated according to three different cell culture methods - 2D cell culture technique (control technique), 3D method using Matrigel®, and 3D method with Hanging-Drop (Fig. 1)”.

Line 206: how long did each passage take? Were the subsequent experiments with the different cultivation experiments only performed in passage 3? I.e. up to P3 the cells were kept in monolayer and only transferred to 3D from P3 onwards?

 To clarify this portion of the text, additional information has been included in various sections of the manuscript:

Lines 153 – 161 “In summary, to establish 3D in vitro culture of YS tissue from different mammalian species, the YS tissues from canine, porcine, and bovine species were first isolated to ob-tain the pool of cells present in YS tissue. Afterward, fresh cell explants from each species were cultured using three different techniques - 2D cell culture technique (section 2.5), 3D culture using Matrigel®, and 3D culture using the Hanging-Drop technique (section 2.3). To determine the best conditions for developing tridimensional structures, eight types of media were evaluated (section 2.4). Once the optimal medium conditions were established, the resulting structures were assessed to analyse morphology and function.”

Lines 176 – 179 “In the Matrigel® technique, fresh YS cells were plated in a 1:25µL ratio of Matrigel® in 48-well plates with 250µL of growth media. Typically, two weeks after the establishment of the initial culture, the Matrigel® culture could be subcultured at a 1:5 ratio every week (Fig 2) until passage 3.”

Lines 190 – 193 - For the Hanging-Drop technique, cells were cultured in 20µL drops of growth media on the lid of a 60mm dish at a concentration of 1:20µL. After the establishment of the ini-tial culture, subculturing occurred every 3-5 days (Fig 3) at a 1:3 ratio until passage 3. Subsequently, morphological and functional analyses were performed.

Line 210: three techniques – you mean 2D, matrigel and hanging drop – please add this information like provided in line 278ff

The passage has been reformulated, and the new phrase can be found on lines 287 – 294: “To track the progress of the cultures over time and compare the three cell culture techniques – 2D culture, 3D Matrigel®, and 3D Hanging-Drop – we utilized the EVOS M500 microscope. The cultures were assessed weekly to perform media evaluation assays and measure the size of the structures generated by each 3D technique. For comparing structure sizes across different 3D techniques, three distinct images of biological replicates were taken for each culture medium and species. These images were captured at 10x magnification and analyzed using ImageJ software [28].”

 Results:

283ff is already the final result – is this based on all examinations?

Lines 287-302 belong to the introduction or discussion. In the results section, there should be a description of your observation. And not just mention the final statement.

The results have been more effectively described in the revised manuscript.

What was the growth of your 3D tissues like in the experimental series - was there a significant difference between the different media, etc.? Provide a table and pictures of the results you obtained from the experiments described in section 2.3 and 2.4, 2.5.

A table detailing the results of various cultures has been added and can be found on lines 391-392.

Additionally, a figure illustrating the outcomes of different cultures has been included on lines 256-257.

Figure 4 is not from each group studied, but only from one trial of each animal. What is the difference between spheroids and organoids? When did you obtain them? Were they part of the same experimental trial? Or were the organoids obtained from matrigel and the spheroids from hanging drop? Please provide more information in the text and figures to understand the method and your final statement.

Fig 5: could you provide more description of the pictures within the pictures? the pictures are also very blurry, so that the structures are not clearly visible

 The descriptions of all figures have been enhanced for better clarity and understanding.

Fig 6 not fully visible - the descriptions in the text can therefore not be understood or reviewed

Fig 7 not fully visible - the descriptions in the text can therefore not be understood or reviewed

 The issued was addressed.

Discussion:

In the introduction, it was mentioned that they are growing these cell cultures to understand miscarriages. Now it seems to be because of diabetic embryopathy and congenital heart disease, tox. testing. Of course there can be more than one target, but within one publication you should stick to one topic.

 Moreover, the further explanation is not a real discussion. It is only a description of the 3D method. Some sentences would rather fit into the method section to provide more detailed information about the manufacturing process.

Furthermore, the authors do not elaborate on their results, but rather state what should have been done further in order to make valid statements.

In my opinion, the discussion should be fundamentally revised, as it does not provide a discussion of the existing results.

The discussion has been substantially revised. In the new manuscript version, the discussion can be found spanning lines 501-559.

Round 2

Reviewer 1 Report (Previous Reviewer 2)

Authors have addressed the issues, I was concerned about. Now is clearly understandable and easy to read. 

Reviewer 3 Report (New Reviewer)

The authors did an excellent and intensive revision of the manuscript. I think this work can be very helpful for the further development of 3D cell culture.

This manuscript is a resubmission of an earlier submission. The following is a list of the peer review reports and author responses from that submission.

Round 1

Reviewer 1 Report

The authors conducted a characterization study to determine the best methodology to culture yolk sac embryonic cells.  It is relevant to identify the most optimal conditions to culture cells and the authors’ conducted a lot of work in an attempt to determine such conditions.  However, justification for the study and the species selected was not well articulated.  The description of the experimental methodologies for a characterization study lacked a lot of detail, which made interpretation and evaluation of the reported results and subsequent discussion difficult to comprehend. Data figures had very poor resolution, tables lacked defined acronyms and excessive use of paragraphs and poor word phrasing throughout the manuscript made it difficult to read.  Some examples of the issues are below:

Grammatic issues: paragraphs, word phrasing is poor, sentences too long:

1.     Excessive paragraphs lines 44-63, lines 80-93, lines 107-115, lines 263-309, lines 311-326. Should be one paragraph

2.     Lines 58-61: Odd wording and sentence is too long.

3.     Line 58: ‘the three greatest reproductive interest species: canine, bovine, and porcine’; comment is overreaching and give no context for the claim.

4.     Line 63: YS exerts??  What does that mean

Section 2.1

1.     How was the 20th-30th day of pregnancy determined?  How do you know the specific age of the embryos? Does specific age of embryo impact culture outcomes?

2.     These were primary cells so were mutated immortalized cells derived from the primary cells?

Section 2.3

1.     Figure 1B: all media components need to be defined in the text.

2.     Figure 2: was this illustration created or borrowed from pre-existing publications? This looks like potential copyright issues.

3.     Table 1: reagents for each culturing methodology?

Section 2.4

1.     What morphological structures were identified and for what purpose?

2.     You dehydrated the tissue and stuck them in a petri dish? No embedding? No storage?

Section 2.5

1.     How was nanodrop utilized to determine RNA purity?

Section 2.6

1.     So embedding did occur?? This section states ‘dewaxing’ (assuming deparaffinization is what was meant) which is not described in section 2.4.

2.     Line 167: What is 20 minutes of ‘rest’.

3.     Line 170: reaction was revealed?

Results

1.     Culture media combinations tested were not actually specified.  Reagents were listed and the companies purchased from but not the actual experimental media conditions.  This is problematic when the purpose of the reported experiment is to determine best culture methods.  Even though a bunch of literature is cited the culture media was untested in current lab where research was conducted prior to testing the 2D vs 3D culture methodologies. It is therefore unclear if culture media conditions would have impacted the outcome or the 2D vs 3D methods.

2.     Figure 2 and 5 resolution is very poor.  Structures are completely unclear.  Pictures should be larger and maybe separate to view more clearly

3.     Completely confused by line 242.  Normalized gene with what specific gene??  None specified.

4.     Figure 6 unfocused data

Reviewer 2 Report

Pereira et al. describe 3D culture of Yolk sac tissue from 3 different species: Porcine, canine and bovine. Using this approach, the authors successfully extracted and cultured the yolk sac tissue in monolayer form as well as 3D culture through 2 different methods. This culture system is definitely relevant and of interest to the field. However, I have major concerns:

The images in the Figures 3, 4 and 5 are very low quality and makes it difficult for the reviewer to interpret any outcomes from it and suggest any further experiment. 

Morphology and cellular conformation: The authors need to further characterize the organoid and spheroid culture system. Even the gene expression patterns are not even close to that of tissue. Authors are requested to find more genes and use qPCR.

Discussion: Authors are advised to rewrite the discussion for the better understanding.